# The Role of *Puroindoline*, *Gpc-B1*, *Starch Synthase* Genes, and Gluten Proteins in Regulating End-Use Quality in Wheat

**DOI:** 10.3390/ijms26178565

**Published:** 2025-09-03

**Authors:** Mantshiuwa C. Lephuthing, Thobeka Philile Khumalo-Mthembu, Toi John Tsilo

**Affiliations:** 1Germplasm Development Department, Agricultural Research Council–Small Grain Institute, Bethlehem 9700, South Africa; lephuthingm@arc.agric.za (M.C.L.); khumalotp@arc.agric.za (T.P.K.-M.); 2Department of Life and Consumer Sciences, University of South Africa, Florida 1710, South Africa; 3Production Systems & Crop Protection, Agricultural Research Council–Small Grain Institute, Bethlehem 9700, South Africa

**Keywords:** genomics, glutenins, grain protein content, proteomics, *puroindolines*, *starch synthase* genes

## Abstract

End-use quality is a crucial aspect of wheat quality, influencing the type and quality of the produced food products. It is mostly determined by the content and characteristics of the protein as well as the starch in the grain. Being a staple food, wheat provides more than 30% of the total calories and proteins in human diets globally. Wheat grain consists of a protein network, called gluten, which provides wheat doughs with their unique viscoelastic properties. The genetic improvement of end-use quality traits is indispensable to meet the requirements of grain markets, millers, and bakers. Thus, modern approaches such as proteomics and genomics are important to precisely identify alleles, genes, as well as their functions in improving end-use quality. End-use quality is mainly regulated by grain protein content, grain hardness and starch synthase genes, as well as gluten proteins, which can be exploited to improve the quality of wheat for the production of desired wheat cultivars. The aim of this review is to highlight the progress, challenges, and opportunities in breeding for end-use quality in wheat. The paper outlines the following key aspects: (1) challenges associated with breeding for end-use quality and (2) opportunities such as genomic selection, marker-assisted selection (MAS), and genetic variation in landraces and wild relatives for end-use quality improvement and the genes regulating end-use quality. Lastly, the paper discusses the prospects for future quality improvement in wheat. The review provides a comprehensive insight into the effects of genes on regulating end-use quality and serves as baseline information for wheat breeders to guide the development and deployment of wheat cultivars for future quality improvement.

## 1. Introduction

There is a rising demand for food with certain quality attributes, driven by rapid population growth coupled with increasing urbanization and related changes in eating habits or food preferences [1,2]. The growing population exerts pressure on researchers to maintain or improve food production expeditiously while improving quality traits to meet the demands of various international market and end users, such as farmers, millers, bakers, and consumers. The change in food preferences for easy and fast food such as bread, biscuits, pasta, and noodles has increased the need for wheat-based products [3]. Subsequently, many wheat breeding programs have focused on increasing grain yield by improving various traits, such as grain yield components, end-use quality, tolerance to abiotic stresses, and disease and pest resistance, to meet the higher standards imposed by the wheat producers, millers, bakers, and consumers [4,5,6]. As a result, notable advancements have been achieved in the past 60 years, with the global wheat yield increasing from 1.2 MT/ha in 1962 to 3.7 MT/ha in 2022 [7].

Wheat (*Triticum* spp. L.) is one of the major crops contributing significantly to human diets, providing about 20% of the total calories and proteins consumed [3,8]. Approximately 95% of the wheat produced globally is hexaploid bread wheat (*T. aestivum* L.), while the majority of the remaining 5% is tetraploid durum wheat (*T. turgidum* L. var. *durum*) [9]. Both bread and durum wheat are important for their nutritional value and are major sources of various components such as carbohydrates, protein, dietary fiber, and essential vitamins that play an important role in the prevention of chronic diseases such as cardiovascular diseases, diabetes, etc. [10,11]. However, these two wheat species differ in their genetic constitution, grain composition, and end-use quality [12,13]. Bread wheat is an allohexaploid species (2*n* = 6*x* = 42, genome AABBDD) and has a genome size of ~17 Gb, one of the largest among crop plants [14]. In perspective, a single bread wheat chromosome is twice the size of the entire rice (*Oryza sativa*) genome (2*n* = 24, genome AA).

Bread wheat flour has unique dough properties, forming gluten protein fractions that make it suitable for bread making. Gluten is a viscoelastic storage protein complex that is essential for producing a wide range of end-use products (such as bread, cakes, cookies, cereals, pastries, pasta, and noodles) in baking factories [3,15]. Durum wheat (2*n* = 4*x* = 28, genome AABB) is a key part of the Mediterranean diet and is used to make pasta. Due to the absence of the D sub-genome, durum wheat has poor bread-making qualities and low grain protein content. Moreover, the presence of a stop codon and/or the insertion of a transposon that silenced the *Glu-A1* locus has deprived durum wheat of significant alleles for bread making [16,17]. Both the quality and quantity of the grain produced determine the value of wheat grain. The grain quality is determined mainly by the grain protein content (GPC) in the endosperm and the texture of the grain [18,19]. End-use quality phenotyping is expensive, as it is time consuming, and a large amount of grain is required to conduct the evaluations. Hence, the selection for end-use quality is often delayed until late in the breeding program.

The end-use quality of wheat is determined by the combination of various related quality traits that are regulated by different metabolic pathways, making the improvement of end-use quality an inherently challenging breeding goal. The complexity is further increased by the fact that many of these traits are also correlated with each other [20,21]. Recently, there has been a shift from phenotyping to more detailed genetic approaches of selecting genes and alleles for desirable traits, mainly wheat grain protein and its subunits, using marker-assisted selection (MAS), quantitative trait locus (QTL) mapping, association mapping, and gene editing technologies [17]. The studies on the functional and molecular genetic basis of wheat quality therefore led to the development of a wide range of wheat products with specific quality attributes [4,22]. High-throughput molecular markers, such as single-nucleotide polymorphisms (SNPs) that are strongly associated with end-use quality traits, provide a means to facilitate a more precise and effective selection in early-generation breeding materials [23]. Despite the large genome size, the advancements in sequencing technologies have enabled researchers to study the wheat plant in its entirety. Moreover, genetic mapping techniques such as quantitative trait locus (QTL) mapping and genome-wide association studies (GWASs) have made it possible to discover the genomic regions and genes linked to different end-use quality traits in wheat. Therefore, knowledge of the molecular basis, genes playing major roles, and their effect on end-use quality is an imperative requirement for modifying the characteristics of end-use traits to enhance the quality for traditional uses and to generate new characteristics for innovative applications. This review discusses the progress made in discovering genes and comprehensive insights into their influences in regulating end-use quality traits, challenges, and opportunities in breeding for improved end-use quality in wheat.

## 2. The End-Use Quality

In wheat, end-use quality is a very complex trait controlled by many genes that interact with each other and the environment. End-use quality comprises many component traits, such as the physical characteristics of the grain, flour composition, the profile of the protein fraction, rheological properties of the dough, and mineral element nutritional quality [12,23,24]. The main traits that define wheat quality are grain protein content, grain hardness, gluten quality, and starch properties [25]. These traits are quantitatively inherited and are regulated by major and minor genes, as well as influenced by genetic effects, the environment, and the genotype-by-environment interaction (GEI) [26,27].

### 2.1. Grain Protein Content

Grain protein content (GPC) is an essential trait that determines the nutritional value, processing, baking properties of wheat for making various food products, and, ultimately, the market value of the grain [23,28,29,30,31]. The GPC is influenced by both the genotype and environmental conditions, such as temperatures and access to water during grain filling, as well as the high levels of nitrogen fertilization [3,32]. Normally, the protein content in the grain ranges between 7% and 8%, with gluten proteins accounting for the majority (~80%) [33]. Higher GPC levels were discovered in landraces and wild relatives than in modern wheat, indicating that landraces and wild relatives could be potential sources to improve GPC. For instance, the wild emmer (*Triticum turgidum* spp. *dicoccoides*) accession, FA15-3, originating from Israel, was identified as a good source of high GPC. This accession is one of the most extensively exploited germplasm to accumulate more than 40% protein content provided that there is sufficient nitrogen [34,35]. The genomic region controlling GPC was mapped on chromosome 6BS and was designated *Gpc-B1* [36,37,38]. Since the discovery of the *Gpc-B1* gene, several research studies have transferred this gene into hard red spring wheat with the success of improved protein content by 3% [38,39,40]. The *Gpc-B1* gene in wheat lines encodes a NAC transcription factor, i.e., *No Apical Meristem* (*NAM-B1*), which accelerates senescence, an important process that increases the distribution and relocation of proteins, nitrogen, and micronutrients [such as iron (Fe) and zinc (Zn)] to the developing grains [38,41]. Different studies have reported positive and significant associations between GPC and micronutrient concentrations, with lines carrying the *Gpc-B1* allele consistently discovered to have higher levels of grain Fe and Zn concentrations [42,43,44]. This indicates that this allele can improve both GPC and micronutrient contents in wheat. Moreover, the functional *Gpc-B1* allele has been introgressed into hexaploid wheat through wide hybridization, and so far, 18 cultivars have been registered and released into different wheat markets [32,45,46].

### 2.2. Storage Proteins

Seed storage proteins are major determinants affecting the end-use quality and production of different wheat-based products. They are categorized into four classes based on their solubility, namely albumin and globulin, which are structural proteins, as well as glutenin and gliadin, the gluten proteins (Figure 1). Structural proteins are considered less functionally relevant than gluten proteins but may play yet undiscovered roles in various biological and technological processes [47]. Gluten proteins are the dominant seed storage proteins found in the starchy endosperm cells of all cereal grains, except for rice (*Oryza sativa* L.) and oats (*Avena sativa* L.) [48,49]. Both monomeric gliadin (singly chained polypeptides) and polymeric glutenin (multiple polypeptide chains linked by disulfide bonds) (Figure 1) constitute more than 80% of wheat flour proteins and are responsible for different rheological properties of the dough [18,50,51,52].

Glutenin proteins are more responsible for the cohesive and elastic properties of the dough and thus play important determining factors of bread-making quality [25,48,53,54,55,56]. They are subdivided into two subunits, the predominant high-molecular-weight glutenin subunits (HMW-GSs), and low-molecular-weight glutenin subunits (LMW-GSs). Although both fractions influence dough strength and extensibility, HMW-GSs contribute about 40–60% of the overall dough quality, despite constituting a small fraction (10%) of the total storage proteins. The LMW-GS is encoded by *Glu-3* homoeologous genes on chromosomes 1AS, 1BS, and 1DS [22]. In addition to extensibility, LMW-GSs have been considered to play a secondary role compared to the HMW-GSs in the regulation of dough properties, and this was due to the difficulty in separating and accurately detecting the LMW-GS alleles. But, in the past decades, approaches to separate proteins and the use of molecular tools have been improved [57,58,59,60,61], allowing many studies to analyze the influence of different LMW-GS alleles on end-use quality. Using diverse sets of wheat materials and approaches, many studies came to a general agreement that both the HWM-GS and LMW-GS play a role in dough and end-use quality traits [33,62], and *Glu-3/Gli-1* alleles can be used to compliment current selection strategies [63].

HMW-GSs are encoded by *Glu-1* genes (*Glu-A1*, *Glu-B1*, and *Glu-D1*) located on the long arms of chromosomes in group 1 (1A, 1B, and 1D) of hexaploid wheat [53,64]. Each of these loci has one x-type and one y-type subunit, theoretically adding to six HMW subunit genes (*1Ax*, *1Ay*, *1Bx*, *1By*, *1Dx*, and *1Dy*) in total. The y-type subunits have been reported to be more valuable than the x-type in improving flour quality and, hence, bread-making quality [46,55]. However, depending on the cultivars, almost all hexaploid wheat cultivars express 3 to 5 HMW-GSs [48,49,65]. The expression levels of *1Dx*, *1Dy*, and *1Bx* subunits are usually the highest in all cultivars, whereas the *1Ax* and *1By* subunits are seldomly expressed. Despite being frequently reported in diploid and tetraploid wheat, the expression of the *1Ay* subunit is mostly silenced in hexaploid wheat cultivars [53,65,66,67].

Gliadins are heterogeneous polypeptide mixtures and comprise approximately 60% of the gluten [33,68]. They are important for wheat end-use quality traits, but because they are encoded by a complex, multigenic family that includes various pseudogenes, the knowledge of the contribution of each gliadin gene is still lacking [69]. They influence dough viscosity and extensibility and are classified into γ-, ω-, α-/β-, and δ-gliadin based on differences in their primary structure [69,70,71,72] (Figure 1). Gliadins are encoded by *Gli-1* genes on 1AS, 1BS, and 1DS, and *Gli-2* genes on 6A, 6B, and 6D chromosomes. The amount and composition of these components affect dough rheology and end-use properties [46,48,52,53,73,74].

### 2.3. Grain Hardness

The grain hardness or texture is an essential component of the wheat grain that distinguishes the market class and trade worldwide [24,75,76]. Grains are classified and graded according to the value of the grain texture using an index [grain hardness index (GHI)] as a criterion [19]. This GHI represents the amount of force or energy required to break the grain. Based on the American Association of Cereal Chemists International (AACCI) method 55-31.01, grain texture ranges from very soft (GHI ≤ 10) to extra hard (GHI > 90) [77]. For example, the United States of America has graded its wheat into three classes: soft, hard, and durum wheat. The classification for the intended wheat end use is of great interest to the farmers, millers, and bakers (i.e., grain traders). Breeders also target the same characteristics to increase grain yield in the field [78].

Earlier research speculated that the distinction between hard and soft wheat was regulated by a single major gene [79], and it was later discovered that the distinction was due to the genetic effect of puroindoline proteins (*Pins*) [80]. *Pins* are small cysteine-rich proteins that are exclusively expressed in the starch endosperm cells of cereal grains [81,82]. Although there are other minor genes, much of the variation of the grain hardness in bread wheat endosperm is regulated by a single locus, the hardness (*Ha*), which is present in the distal end of the short arm of chromosome 5D [83,84,85,86]. In *Triticeae* diploids, the *Ha* locus is found on the group 5 chromosomes, and the locus consists of a group of three major genes, i.e., *Pina-D1*, *Pinb-D1*, and *Gsp-1* genes, which encode for *Puroindolines a* (*Pina*), *Puroindolines b* (*Pinb*), and *Grain Softness Protein-1* (*Gsp-1*), respectively [74,86,87]. All three *Gsp-1* genes from the A, B, and D genomes are preserved in bread wheat, but their roles remain unclear [86,88]. The *T. aestivum* group exhibits textures that range from soft to hard. When both *Pina* and *Pinb* (major polypeptides) are functional, the texture of the endosperm is considered soft; and when one protein is mutated, absent, or nonfunctional, the texture is hard, as in the case of durum wheat [33,85]. Durum wheat is a contributor of the A and B genomes of bread wheat. During the evolution, both *Pin* genes were deleted from chromosomes 5A and 5B of durum wheat. Thus, the *Ha* locus is only present on chromosome 5D in bread wheat. To date, 9 and 17 *Pina* and *Pinb* alleles, respectively, have been discovered, of which the major alleles detected in hard wheat cultivars are *Pina-D1b*, *Pinb-D1b*, *Pinb-D1c*, and *Pinb-D1d* [86,89]. *Pina-D1*-related genes were discovered in other crops such as oat, rye (*Secale cereal*), and barley (*Hordeum vulgare* L.), and *Pinb-D1*-related genes were found in oat and barley [86,90].

### 2.4. Starch Properties

Wheat starch is the main storage carbohydrate representing approximately 70% of the dry matter of the endosperm [91,92]. It is an important by-product of gluten production, used as the main source in bread, noodles, and cookies [48]. Being an essential part of the food industry, starch is widely used in food, paper, textile, chemical, and pharmaceutical industries as a thickener, stabilizer, adhesive, and as agents (gelling, water-retaining, and bulking) [93]. During the grain-filling stage in wheat, starch accumulation peaks between 12 and 35 days after anthesis [94,95]. Starch contributes significantly to creating high-yielding wheat cultivars because the improvements in grain size attained through breeding are mostly due to increases in starch content.

Starch granules consist of two major components of glucose polymers called amylose (resistant starches) and amylopectin (waxy starches) and are largely determined by the ratio of 1:3 [96]. A higher amylose ratio in starch is of particular interest because it contributes to resistant starch (RS) in food, which is not easily broken down by digestive enzymes in the body and is beneficial to human health [97]. The starch granules are defined based on D-glucose residue, and they have a unique trimodal distribution (A-, B-, and C-type), each differentiated by its properties [98,99,100,101]. The amylose and amylopectin polymers are synthesized by granule-bound starch synthase (GBSSI) or waxy protein in the amyloplast, and they differ in structure and properties [102,103,104,105,106]. Amylose is a relatively long linear α-glucan formed by α-(1,4) residue, and it represents 22–35%, while amylopectin is a heavily branched structure consisting of approximately 95% α-(1,4) residues ramified every 20–30 residues by 5% α-(1,6) linkages representing 65–78% [92,105,107,108]. The interaction between the polymers might have an impact on the physical and chemical properties of starch (gelatinization, pasting, and gelation), as well as the quality of the end products. Starch properties significantly influence wheat flour or semolina-based food products. For instance, in the production of Asian noodles, wheat with a low amylose level is preferred because it improves starch viscosity and flour swelling volume [109,110,111]. Moreover, starch is associated with the nutritional value and the shelf life of pre-cooked products.

Various end-use quality traits are influenced by the effects of major genes. The genetic architecture of grain hardness, for example, is primarily controlled by the *puroindolines*, gluten strength by the high-molecular-weight glutenins, and starch paste viscosity by the *granule-bound starch synthase* (‘waxy’) genes [112,113]. However, because of parent selection or early-generation phenotypic and/or genotypic selection, these major genes are frequently fixed in elite breeding populations and do not adequately account for the levels of end-use quality required for cultivar release nor for the range of variation observed among breeding populations [114].

## 3. Genetic Control of End-Use Quality Traits

Genetic improvement is the root of crop breeding. Therefore, it is crucial to understand each quality trait’s heritability, genetic basis, and the extent to which their variation is influenced by various environmental circumstances. Genes with major influence on the end-use quality of wheat include *Gpc*, *Ha*, *SS*, and the HMW/LMW-GS [41,53,115,116] (Table 1).

### 3.1. The Influence of Gpc-B1 Genes on Grain Protein Content and Nutritional Quality

Among the end-use quality traits, GPC has received special attention, as it is an indicator of the quality performance of wheat products. According to several studies, QTL associated with GPC were detected in all wheat chromosomes [131,132,133,134]. Of particular importance is the GPC locus on chromosome 6B of bread wheat. This locus harbors the *Gpc-B1* gene, which was introgressed from the wild emmer (*Triticum turgidum* L. ssp. *dicoccoides*) wheat into the bread wheat cultivars, thereby improving the content of grain protein [32,41,135,136]. In addition, the *Gpc-B1* functional allele introgression had a significant effect on the soft durum wheat grain, thereby increasing most of the grain and flour protein, dough mixing strength, and bread-making traits, with fewer effects on milling performance [137]. Ohm et al. [138] reported that an increased grain protein content linked with the *Gpc-B1* alleles was found associated with an increase in storage proteins (HMW and LMW) and gliadins (α-, β-, γ-, and ω-), both in the SDS soluble and insoluble fractions [138]. The presence of the functional *Gpc-B1* allele in plants showed an increase in the wet gluten, longer mixograph mixing time and peak height, higher Zeleny sedimentation volume, and improved spaghetti quality across three contrasting environments [135,139]. Moreover, the GPC and micronutrient content were discovered to be positively correlated, and all lines carrying the wild-type *Gpc-B1* allele were consistently observed to have significantly increased Fe and Zn concentrations [38,41] (Table 1). Overall, the introgression of the *Gpc-B1* functional allele into wheat has many advantages and can improve grain and flour protein, dough strength, and bread-making quality.

### 3.2. The Influence of Glu-1, Glu-3, and Gli-1 Genes on the End-Use Quality of Wheat

Gluten, a major wheat storage protein, is a coherent mass formed when glutenin and gliadin (storage proteins) bind after water is added to flour. As mentioned above, the HMW-GSs are encoded by the homoeologous genes at the *Glu-1* loci (*Glu-A1*, *Glu-B1*, and *Glu-D1*), while LMW-GS proteins are encoded by a multigene family located at the *Glu-3* loci (*Glu-A3*, *Glu-B3*, and *Glu-D3*) [53,140,141]. Changes in wheat functionality have been linked to allelic variation at each of these loci. Most of the studies conducted comprised of quite a limited number of genotypes and it was unclear which genomic region played a major role in the dough properties. Recently, in a study using the quality data of 4623 grain samples, generated across 10 years at the CIMMYT bread wheat breeding program, the glutenin alleles (*Glu-A1a*, *Glu-A1b*, *Glu-B1al*, *Glu- B1i*, *Glu-B1f*, *Glu-D1d*, *Glu-A3b*, *Glu-A3d*, *Glu-A3f*, *Glu-B3c*, and *Glu-B3d*) were significantly associated with stronger gluten strength, good extensibility, and higher bread loaf volume [33]. This demonstrated that gluten strength is strongly influenced by the combination of glutenin variations (both HMW-GSs and LMW-GSs), with *Glu-B1*, *Glu-D1*, and *Glu-B3* loci having the greatest effect.

The *Glu-1* genes largely control the dough rheology properties of wheat, which are estimated by water absorption, dough development time, dough stability, maximum dough resistance, dough extensibility, and flour paste viscosity [142,143]. These dough rheological properties, as well as other traits like flour protein content, particle size, loaf volume, and crumb score, can be used to estimate wheat baking quality [144]. The HMW-GS loci *Glu-A1* and *Glu-D1* in particular have been shown to improve the microstructures and aggregation of gluten matrix, resulting in excellent rheological properties of wheat dough [145,146,147,148]. (The *Glu-1* allele encoding *1Ax* and *1Ay* subunits (HMW-GSs) was successfully introduced into two Australian wheat cultivars and the introgression improved protein, gluten contents, and the bread-making properties, without negative effects on the agronomic traits [56,149]). Wang et al. [46] also reported that the TaAy7-40 line containing the active *Glu-1Ay* allele displaying increased grain protein content, better processing quality, improved grain weight, and increased grain size. This validated the notion that the y-type subunits contribute more than the x-type in improving bread-making quality.

Due to the lack of the D genome and, thus, the absence of the *Glu-D1* and *Glu-D3* proteins, durum wheat is considered to have lower gluten strength than bread wheat [150]. Recently, the alleles corresponding to the HMW-GSs Dx2 + Dy12 (*Glu-D1a*) and Dx5 + Dy10 (*Glu-D1d*) were introduced into durum wheat, and they improved SDS sedimentation volume, lactic acid solvent retention capacity, and mixograph dough mixing parameters, thereby increasing dough strength and bread-making quality [138,151]. Wesley et al. [152] evaluated the effects of genetic variation in wheat glutenin and gliadin protein alleles on dough mixing characteristics and bread and noodle quality, and both gliadin and glutenin were found to influence wheat flour properties for making bread and noodles. The study hypothesized that variations at the *Glu-3/Gli-1* loci could explain variation in bread and noodle production.

### 3.3. The Influence of Pina and Pinb Genes on the End-Use Quality of Wheat

The texture or hardness of the wheat endosperm is an important factor that determines the technological and end-use quality, as it affects both milling and baking properties of wheat. The grain texture is regulated by the hardness locus (*Ha*), which is controlled by *puroindoline* (*Pina* and *Pinb*) genes. Additionally, *Pin* genes control the composition of proteins and different antimicrobial activities [153]. The effect of *Pin* genes on quality traits was evaluated using a diverse germplasm of wheat genotypes, and the genotypes with *Pina-D1b/Pinb-D1b* double-mutation *pin*-genes (hard texture) showed the highest grain protein content, thousand kernel weight (TKW), and SDS-sedimentation value (positive association). However, the ash content had a negative association with the *Pina-D1b/Pinb-D1b* double mutant [154]. In contrast, genotypes with *Pina-D1a/Pinb-D1b* exhibited improved extensibility, dough development time, and milling yield compared to the *Pina-D1b/Pinb-D1a* genotypes [155,156]. When examined in vitro or expressed in transgenic plants, *Pin* proteins had inhibitory effects on plant pathogenic fungi [81,157,158,159]. Moreover, *Pins* were found to have antimicrobial activity, suggesting that these proteins might be employed as food preservatives in baking products [153]. A *Pina*-overexpressing line was crossed with a *1Ax-1*-overexpressing transgenic durum line, and the lines shared the same genetic background. The results indicated the combining effects on dough mixing parameters when transgenic *Pina* and *1Ax-1* are stacked in durum wheat [160]. Similarly, in a study using transgenic lines of durum wheat, a synergistic or additive effect of *Pina* and *1Ax-1* on viscosity was reported, suggesting that *Pins* affect several food-processing qualities, such as dough mixing and pasting quality, by interacting with gluten proteins [148]. Moreover, interactions between glutenins and *Pinb* alleles significantly influence the polymer characteristics and percentages of ω-gliadins [161]. These reports demonstrate the significance of gene stacking through transgenic approaches for quality improvement in durum wheat with dual purpose (for pasta and bread). Despite the enigmatic nature of puroindoline proteins, they undoubtedly play a crucial role in wheat.

### 3.4. The Influence of GBSSI (Waxy) Genes on the End-Use Quality of Wheat

During grain filling, four isoforms of starch synthase (*SSI*, *SSII*, *SSIII*, and *GBSS*) are expressed in the endosperm of cereal crops [162]. Each isoform contributes differently to overall starch synthase activity, and gene-specific mRNA levels can be used to predict the presence of enzymes. The three soluble SSs (*SSI*, *SSII*, and *SSIII*) are responsible for the structure and size of the amylopectin clusters, while *GBSS* is a key enzyme in the synthesis of amylose in starch granules [124,126]. There are two types of *GBSS* genes, *GBSSI* and *GBSSII*, in cereals such as *T. aestivum* L., barley, maize (*Zea mays* L.), and rice [163]. However, in *T. aestivum*, the *GBSSI* (*waxy*, *Wx* protein) is the most abundant protein inside the starch granules. It is responsible for the synthesis of long amylose glucan chains stored in endosperm and pollen [164] and plays a role in the synthesis of long chains of amylopectin [162,165,166,167]. There are three homoeologous *waxy* genes (*GBSSI*) (*Wx-A1*, *Wx-B1*, and *Wx-D1*) located on each of the genomes that encode the three *GBSSI* isoforms on chromosome 7AS, 4AL, and 7DS, respectively [168]. The *Waxy* gene originally present on chromosome 7B was translocated to chromosome 4AL; this process occurred during the evolution of wheat, resulting in an exchange of genetic materials between chromosomes 7B and 4A [169,170]. The three homologous *Wx* genes do not contribute equally to amylose biosynthesis, but their interaction determines the amylose content of wheat starch [164]. According to Graybosch [171] and Geera et al. [164], mutations in the gene responsible for one or two *GBSS* null or non-functional alleles result in starch with a decreased amylose content (termed as partly waxy starch), while mutations in the gene encoding three *GBSS* null or non-functional alleles produce virtually amylose-free or waxy starch. In the absence of the GBSS enzyme, grain endosperm tissue consists almost entirely of amylopectin [172]. Similarly, mutations in amylopectin synthesis genes, such as *SS* or *starch branching enzyme* (*SBE*) genes, result in starch synthesis with a higher amylose content (RS) [25]. Certain Asian wet noodle products use flours with optimal quality attributes sourced from partially waxy wheats. Furthermore, the evolution of waxy wheat with respectable agronomic performance depends on partly waxy wheat. Waxy wheat flour can also be used to increase the shelf life of baked products without affecting the gluten content of the wheat.

## 4. Challenges in Breeding for Improved End-Use Quality

### 4.1. Quality Deterioration

Quality deterioration is one of the major attributes negatively affecting end-use quality traits and thus requires consideration during the improvement of wheat cultivars. Naturally, end-use quality tends to deteriorate with time due to factors such as environmental conditions, storage conditions, pest and disease infestations, as well as alien introgression [17,173]. Although deterioration is inevitable, it can be managed by selecting wheat seed lots with high vigor. Seed vigor is the primary index used to measure seed quality in wheat, and seeds with high vigor have been shown to exhibit an extended lifespan or longevity and thus deteriorate slowly than seeds of low vigor [174,175,176,177].

Diseases infestation such as foliar diseases impacts the rheological properties of wheat dough, the physical quality (hectoliter weight) of grain, and the quality of baked goods [178,179,180,181]. Stripe rust (*Puccinia striiformis* f. sp. *tritici*) is one of the important fungal diseases of wheat worldwide deteriorating the milling quality. Stripe rust significantly reduces flour yield (FY) and increases flour ash content (FAC) due to shriveled seeds, consequently affecting seed quality and grain yield [182,183,184]. Weak correlations between stripe rust and dough or bread-making traits have previously been reported, suggesting that stripe rust does not deteriorate much of dough and bread-making quality [185]. However, more studies are required to clearly elucidate the influence of stripe rust infection on grain protein content and end-use quality.

The introduction of novel and useful genes from various gene pools plays a significant role in improving wheat productivity. Alien introgression has many advantages related to an increase in resistance to abiotic and biotic factors and has proven to provide new sources of alleles and genes. In contrast, it significantly reduces the baking quality. The introgression of large alien chromosomes poses challenges, for instance, causing the linkage drag that often contributes negatively to the agricultural value of the wheat line [185,186,187]. To alleviate this challenge, genes of interest can be introgressed into the wheat genome instead of the chromosome to avoid the linkage drag. However, such transfers are usually blocked by the presence of a major pairing homoeologous gene (*Ph1*) in the long arm of chromosome 5B. The *Ph1* allele ensures strict control of homologous pairing across the hexaploid genome while preventing homoeologous pairing between wheat and an alien species [188,189]. With improvements in high-throughput genotyping platforms and phenotyping tools, progress has been made to transfer genes of interest with small, desired alien chromosome segments from wild species to wheat with reduced linkage drag. For example, one method that has been extensively used for manipulation of homoeologous recombination to improve crops is the deletion of the *Ph1* gene in the mutant stock *Ph1b*, which permits homoeologous pairing to occur, allowing for limited gene transfer [190,191,192,193,194,195].

The most successful alien transfer into the wheat genome is the short arm of chromosome 1R translocations, originating from the secalin storage protein of rye (*Secale cereale* L.) crop [196]. The 1RS carries important genes conferring resistance against fungal pathogens such as the *Yr*9 gene (stripe rust), *Lr*26 gene (leaf rust), *Sr*31 gene (stem rust), and *Pm*8 gene (against powdery mildew) in wheat breeding [191,197,198,199,200]. The 1RS.BL also has the *Dn7* gene that provides broad-spectrum resistance to various Russian wheat aphid (RWA), *Diuraphis noxia*, biotypes [201]. The RWA is a global pest of small grains that significantly affect wheat production [202]. Despite positive effects of 1RS/1BL loci on traits such as diseases, pests, root biomass, spikes per square meter, grain and test weight, grain protein content, and drought tolerance, variation from significant to non-significant effects on grain yield were detected [203,204,205], while other studies reported high grain yield even under water-stress conditions [206,207].

Unfortunately, the 1RS.1BL is known for deteriorating bread-making quality because of the introduction of the *Sec-1* locus and the loss of *Glu-B3/Gli-B1* on 1BS, thereby resulting in weaker gluten and less tolerance to overmixing of dough [17,22,208,209,210] (Table 2). The 1RS carries the *Sec-1* locus, which encodes high-molecular-weight proteins γ- and ω-secalin [186,211]. These proteins are the main cause of celiac disease, an autoimmune disease that causes inflammation of the small intestine [212]. The 1RS.1BL translocation with *Glu-B3j* (null) allele, a well-known region having a large negative effect on quality, was involved in 10 quality parameters [160,209,213]. To mitigate these quality problems, the 1RS arm was engineered using induced homoeologous recombination [214]. The process involved the elimination (rye *Sec-1* locus) and substitution (wheat *Glu-B3/Gli-B1* loci) of two interstitial rye segments with wheat chromatin [209]. In contrast, Hysing et al. [215] reported that the introgression of rye from 2BS.2RL wheat–rye translocations might have minor effects on the baking quality due to the insignificant differences between translocation and non-translocation groups for traits such as grain protein content, starch, alpha-amylase activity, and other agronomic traits.

### 4.2. The Effects of External Factors on Wheat Grain Quality

Wheat is a high-yielding cereal crop, and over the past few decades, significant improvement in the production has been achieved. However, the environment plays a significant role in determining the quality of wheat grain. The genotype, environmental conditions, and the GEI interaction influence end-use quality traits. For example, the GEI influences polymers and changes the formation of grain starch and protein [223]. To improve end-use quality traits using marker selection, a clear understanding of the genetics of the desired trait as well as the environmental influence is crucial. Due to the predicted climate change, the world is experiencing recurring incidents of unexpected fluctuations in temperatures, and abiotic stresses (e.g., recurrent droughts, high temperatures, etc.) occur frequently and simultaneously, which all negatively influence wheat productivity and grain quality. Thus far, the results reported have depicted that abiotic stresses trigger complex proteomic changes in wheat grains, affecting the expression of the proteins and starch accumulation due to the sensitivity of their regulation to abiotic stresses.

All growth stages of wheat are sensitive to high temperatures, but the reproductive stage is the most critical. During the reproductive stage, the heat stress predominantly accelerates the senescence rate, decreases the grain-filling period, and subsequently results in reduced grain weight, defragmented starch granules, and reduced overall seed quality [224,225]. However, the accelerated senescence rate was reported to increase the distribution and relocation of proteins, nitrogen, and minerals such as Fe and Zn to the developing grains [32,41,226]. In a study conducted in different locations, the grains from an environment affected by severe drought and heat stress were small and shriveled, thus exhibiting higher grain Fe and Zn contents and lower grain yield components [227]. These studies demonstrate the challenge of improving grain yield together with grain protein content and nutritional quality due to the dilution effect. During the flowering or post-anthesis period, drought and heat stresses tend to enhance the accumulation of α- and ω-gliadins and HMW-GSs, though differential effects have been reported on the accumulation of different LMW-GSs under abiotic stress [225,228,229,230,231,232]. The effects of day and day–night combined heat stresses were investigated during the grain-filling stage using gene expression and proteomics approaches. The heat stress downregulated the HMW-GS proteins, while the LMW-GS α/β- and γ-gliadin proteins were upregulated [116]. Similarly, reduced synthesis of glutenins and stable or increased synthesis of gliadins under heat stress were reported, which might be due to the synthesis of gliadins and LMW-GSs early during grain development as compared with HMW-GSs [233,234]. In contrast, reduced glutenin/gliadin and HMW/LMW ratios were observed, which consequently decreased the baking quality of wheat [235]. The findings indicated that the genotypes, types of stresses, and growth stages when stress was encountered had significant influence on the changes in gluten proteins.

Starch properties are more sensitive to high temperatures and are negatively influenced by abiotic stress during plant growth (especially the reproductive stage) [236,237]. High temperatures and drought influence the expression of genes that encode enzymes involved in starch biosynthesis [238]. During the grain-filling period, these abiotic stresses restrict and reduce the accumulation of starch and modify the size of distribution of starch granules in grains due to decreases in the activities of starch metabolism enzymes [239,240]. Moreover, the heat stress decreased the expression of the transcription factors (*TaRSR1* and *OsbZIP58*) that regulate starch biosynthesis [116]. Consequently, the starch content deposition and total grain yield are decreased due to the reduction of the functions of metabolism enzymes, together with their genes responsible for converting sucrose to starch [241,242].

## 5. Opportunities for End-Use Quality Breeding

### 5.1. Genetic Variation as a Source of End-Use Quality Traits

Genetic variation is a prerequisite for the initiation of crop improvement programs. Many studies have been conducted with the aim of studying the genomic regions governing the genetic variation for quality traits, such as grain protein content, grain hardness, flour yield, carbohydrates, micronutrients, and many more [38,132,243]. However, the end-use quality traits of the present wheat cultivars show narrow genetic variation, suggesting that beneficial allelic variation might have been lost due to genetics, bottlenecks, and the Green Revolution’s replacement of landraces with high-yielding modern cultivars. Natural variability within a population changes over time and space because of the interplay of several evolutionary mechanisms, including natural selection, artificial selection, mutations, gene flow, and genetic drift [244]. Various ways such as developing segregating materials through nurseries, hybridization, and mutation breeding can be used to enhance genetic variation [245].

To broaden genetic diversity or introduce new traits to an already-existing breeding program, introgression strategies to transfer single or multiple favorable alleles from landraces, wild relatives, or other germplasms have been employed. These strategies are important to mine for novel allelic variations to expand the genetic basis of modern wheat cultivars, and they reduce the time required to create an improved variety and the issue of linkage drag [17,246,247] (Table 2). Different germplasm banks are currently being explored to identify novel variations among the vast wheat genetic resources. For example, with over 10,000 accessions, the CIMMYT’s wheat germplasm bank collection is continuously being characterized using various techniques to potentially utilize its unexploited variation for the genetic advancement of the breeding program [78,248,249,250].

One example of introgression that has been widely explored is the *Glu-A1* locus, which was suggested to have originated from the wild progenitor of durum wheat *T. dicoccoides* [251] The authors evaluated the *Glu-A1* locus in a Swedish bread wheat line (W3879) and discovered it to express active *1Ax* and *1Ay* alleles, designated as *1Ax21* and *1Ay21** subunits, respectively. The line was utilized, using four cycles of backcrosses, to introgress the active *1Ay* allele into an Italian bread wheat line. Later, Roy et al. [154,216] (Table 2) introgressed the *1AY21** allele into the Australian bread wheat variety, Lincoln, by a backcrossing and selfing scheme, replacing the silent *1Ay* HMW-GS allele. The *1AY21** allele improved the storage protein composition, protein content, and bread-making performance without grain yield penalties or impacting other agronomic traits. Moreover, Rogers et al. [252] introgressed two alleles (*Glu-A1r* and *Glu-A1s*) from *T. boeoticum* Boiss ssp. *thaoudar*, encoding the x-type and y-type subunits, into bread wheat cv. Sicco. The alleles improved the gluten strength predicted by the SDS-sedimentation test, improved stability during mixing, and reduced dough stickiness. In another study, the effects of *1RS.1BL* translocation were studied using a doubled haploid population. Although the translocated genotypes accumulated more protein in the endosperm than non-translocated genotypes, the *1RS.1BL* translocation resulted in a reduction in gelatinization of starch and a reduction in the elasticity, tenacity, and strength of the dough, and the tolerance to overmixing was significantly lower in translocated genotypes [217]. Introgression of the 1E-encoded storage protein from *Agropyron elongatum* also enhanced the bread-making property of Chinese Spring wheat [219] (Table 2).

The landraces and wild species (wild einkorn) were found to have higher allelic diversity of *Pina* genes compared to cultivars, indicating that they could be promising sources for *Pina* genetic variability and potentially help improve grain texture in wheat [87,115,250,253,254]. Moreover, the soft-durum wheat germplasm has been established by introgression of the *Ha* locus into diverse durum wheat varieties [220,221]. This major advance has extended the culinary use of durum wheat and thus transformed the way durum wheat grains are used in the industry.

### 5.2. Marker-Assisted, Genomic, and Phenomic Selection

The advances in sequencing technologies have made it possible for researchers to study the wheat genome in its entirety. Modern breeding techniques, such as marker-assisted selection (MAS) and genomic selection (GS), enable accurate and efficient prediction of quality attributes, which speeds up crop improvement and cultivar development efforts [24,92]. MAS is a process of selecting individuals based on trait-linked markers, and it has been used to improve many traits, including quality traits such as kernel texture, grain protein content, and starch [26,255]. In contrast, GS is a promising strategy utilized to estimate the genetic value in order to select favorable candidates based on the genomic estimated breeding value (GEBV) predicted from genome-wide markers and performance records. It was first introduced in animal breeding by Meuwissen et al. [256] and has been widely adopted worldwide due to its capacity to enhance genetic gains, decrease phenotyping, shorten cycle times, and improve selection accuracy [257,258,259]. It is an effective technique for assisting early generation selection of complex traits such as yield and disease resistance in wheat [260,261,262], as well as recently for wheat processing and end-use quality traits [263,264,265].

Marker-assisted selection and GS techniques have considerably shortened the development time for new crop varieties. For genetic mapping, various molecular markers such as restriction fragment length polymorphism (RFLP), simple sequence repeats (SSRs), single-nucleotide polymorphism (SNP), and Kompetitive Allele Specific PCR (KASP) are utilized to detect genomic regions or markers associated with the trait of interest [266,267,268] to improve quality traits in early generations of the breeding program [131,133,269,270,271]. The QTL mapping and GWAS methods have assisted researchers to dissect the genetic architecture regulating the variability of complex traits and identify loci/genes associated with various end-use quality traits in wheat (Table 3). The QTL regulating protein content in hexaploid and wild wheat have been located on all 21 wheat chromosomes. However, most of them are minor QTL and unstable when studied in different environments [31,132,272,273].

Although the study of genomics has advanced significantly in recent decades, allowing scientists to sequence and analyze complete genomes, understanding the complex relationship between genes and their expression in phenotypes requires a better understanding of plant phenomics [281]. Plant phenomics offers an alternative strategy that can be used to address the high costs of labor and time associated with traditional phenotyping. The strategy utilizes high-throughput phenotyping approaches to quickly evaluate characteristics and enhance the yields, disease resistance, etc., of crops, thereby offering a potential solution to the delayed evaluation and selection for end-use quality traits [282].

### 5.3. Genome Editing Technologies

Genome editing technologies refer to the process of deleting, inserting, or substituting genes or mutation of a DNA sequence at specific target region in the genomes of many crops. These technologies can precisely target any gene of interest using different sequence-specific nucleases [15,283] such as zinc-finger nucleases (ZFNs), transcription activator-like effector nucleases (TALENs), and Clustered Regularly Interspaced Short Palindromic Repeats/CRISPR-associated protein (CRISPR/Cas9) [284]. Recently, CRISPR/Cas9 has become an increasingly successful technique for plant research and crop genetic improvement in wheat and other crops because it is cost effective, highly efficient, and has strong reproducibility [285,286,287,288]. The use of genome editing has enabled the improvement of various crop plants including wheat, maize, and *Arabidopsis thaliana* [285,289,290]. In hexaploid wheat, important traits such as increased powdery mildew resistance [291], grain size, and weight [292,293,294,295] and improved tolerance to pre-harvest sprouting [296] and drought tolerance [297] have been successfully improved through CRISPR/Cas9. To date, various end-use quality traits have been successfully improved. For example, Zhang et al. [294] investigated the gene editing mutants that lack one, two, or all three (*A1*, *B1*, and *D1*) homoeologs of *TaGW2*, and found that these homoeologs increased the grain protein content in lines with double or triple mutants. In another study, low-gluten wheat was developed using two sgRNAs designed to target the conserved region of *a-gliadin* genes, which reduced the gliadin content (i.e., allergenicity) by 85% [298]. Recently, Li et al. [299] used targeted mutagenesis of a gene involved in starch synthase, *TaSBEIIa*, through CRISPR/Cas9 in wheat cultivars Zhengmai 7689 and Bobwhite. The authors successfully produced a high-amylose wheat with significantly increased resistant starch, amylose, protein, and soluble pentosan contents, which presents an opportunity for a significant improvement in human health.

Genome editing technologies revolutionized the plant research field and hold promise to advance wheat improvement. Although CRISPR/Cas9 faces several drawbacks including efficient cellular delivery, off-target effects, immunological reactions, optimization of editing efficiency, and ethical concerns [300], it has a great potential and provides promising avenues for the development of wheat cultivars that are climate resilient, have enhanced nutritional value, and have better processing qualities [301]. Moreover, genome editing provides great amounts of genetic information through the integration of technologies such as trait dissection, speed breeding, GWASs, gene discovery, and editing.

## 6. Outlook and Conclusions

Bread wheat is the key ingredient of cereal-based processed foods including bread, noodles, and cookies. Over the past decades, since the Green Revolution and the introduction of marker-assisted breeding, wheat production has improved significantly, but wheat is still facing unprecedented obstacles due to the changing climate, growing global population, and water scarcity in arid and semi-arid regions [1]. In wheat breeding programs, the analysis of end-use quality is an essential component. However, the high expense and a large amount of grain required at early development stages, as well as the environment and genotype-by-environment interaction, often make testing and evaluating end-use quality traits difficult. Gaining knowledge of the intricate genetic basis of grain end-use quality traits and identifying molecular markers linked to traits of interest for marker-assisted selections can assist breeders in creating cultivars with improved end-use quality effectively. The identification of QTL associated with end-use quality has received a great deal of attention. Unfortunately, the genetic and genomic bases of end-use quality in wheat are poorly understood. The advances in molecular marker technologies and the availability of wheat sequence reference genome, which enabled the annotation of functional genes, have improved the discovery and understanding of the genome architecture and gene expression and provided an opportunity to reduce the breeding cycle, thereby accelerating the genetic selections for significant breeding traits.

The different genes, *Gpc-B1*, *Pina*, *Pinb*, *SS*, *Glu-1*, and *Gli-1*, play an important role in regulating the end-use quality of wheat. Genomic selection for quality traits at an early stage in wheat breeding has been made possible by the introduction of molecular markers associated with quality traits, which has increased the rate of genetic gain in wheat quality breeding. The exploitation of wheat ancestors, landraces and wild relatives through translational genomics will lead to the discovery of the wealth of various important alleles due to their existing rich variability, which can be used as prospective parent material for future wheat breeding initiatives aimed at enhancing quality in modern wheat cultivars. Endeavors are underway to ascertain valuable new variants from the extensive wheat genetic resources housed in germplasm banks. This information could be helpful for modern breeding projects to create materials with novel qualities.

## Figures and Tables

**Figure 1 ijms-26-08565-f001:**
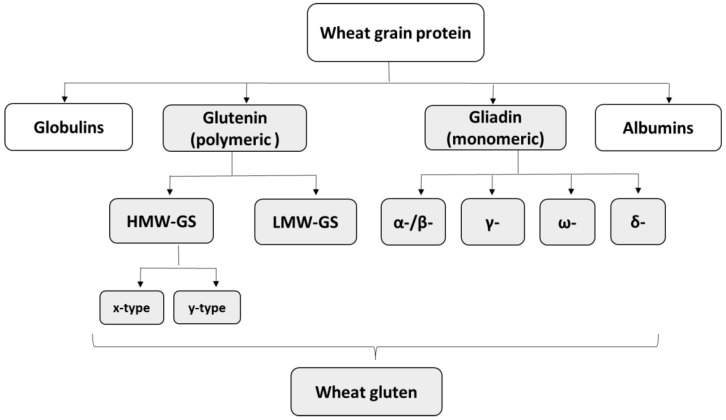
Classification and nomenclature of wheat grain proteins [17].

**Table 1 ijms-26-08565-t001:** Genes regulating wheat end-use quality traits.

Trait	Protein/Enzyme	Genes	Chromosomes	References
Grain protein content	Protein	*Gpc-B1*	6BS	[38,41]
Grain hardness	Puroindoline a	*Pina-D1*	5DS	[85]
	Puroindoline b	*Pinb-D1*	5DS	[117]
Gluten quality	HMW-GS	*Glu-A1*	1AL	[22]
		*Glu-B1*	1BL	[73]
		*Glu-D1*	1DL	[73]
	LMW-GS	*Glu-A3*	1AS	[22,60]
		*Glu-B3*	1BS	[60]
		*Glu-D3*	1DS	[118]
	γ and ω-gliadins	*Gli-1*	1AS, 1BS, 1DS	[69,119]
	α/β-gliadins	*Gli-2*	6AL, 6BL, 6DL	[120,121]
Starch properties	GBSSI or waxy	*Wx-A1*	7AS	[106,122,123]
	GBSSI or waxy	*Wx-B1*	4AL	[123,124]
	GBSSI or waxy	*Wx-D1*	7DS	[123,124]
	Starch synthase I	*SSI*	7AS, 7BS, 7DS	[125,126]
	Starch synthase IIa	*SSIIa*	7AS, 7BS, 7DS	[127,128]
	Starch synthase III	*SSIII*	1AS, 1BS, 1DS	[129,130]

**Table 2 ijms-26-08565-t002:** Introgressions from wild relatives to improve wheat end-use quality traits.

Wheat Type	Introgressed Allele/Genes/Protein/Locus	Source	Protein Type	Characteristics	Reference
Bread	*1Ay21**	*T. dicoccoides* or *T. urartu*	HMW-GS	Improved protein content, storage protein composition, and bread-making quality	[149,216]
Bread	*1RS.1BL*	*Secale cereale*	Storage protein secalin	Deteriorates bread-making quality	[208,210,217]
Bread	*1E*	*Agropyron elongatum*	Storage protein	Enhanced bread-making quality	[218]
Durum	*Glu-1D locus*	*T. aestivum*	HMW-GS	Positive effect on bread-making quality	[219]
Durum	*Ha locus* (*Pin-D1* genes)	*T. aestivum*	Hardness	Improved the grain texture in durum wheat	[220,221]
Bread	*Glu-1Ey*	*Thinopyrum elongatum*	HMW-GS	Improve grain protein content, wet-gluten content, flour, and bread volume value	[222]

**Table 3 ijms-26-08565-t003:** Functional markers reported for end-use quality traits in wheat cultivars for allele identification (adapted from [92]).

Trait	Gene/Locus	Marker	Allele	Cultivar/ Accession	Reference
Protein content	*Gpc-B1*	SSR	Gene specific	Langdon	[38,41]
Grain hardness	*Pina-D1*	STS	*Pina-D1a*, *b*	Chinese Spring, Zhongyou 9507	[243]
	*Pinb-D1*	STS, CAPS	*Pinb-D1a*, *b*, *c*, *d*, *e*, *p*	Chinese Spring, Lorvin10	[248,274]
	*Pinb-B2*	Pinb-B2v2 (Pinb2_IND)	*Pinb-B2a*, *b*	Chinese Spring, Zhongmai 175	[275]
HMW-GS	*Glu-A1*	KASP	Ax1, Ax2 a, AxNull	Chinese Spring (CS), Opata 85	[276,277]
	*Glu-B1*	STS, KASP	Bx7, 8, 9, 13, 14, 15, 16, 17, 20, 23	Various markers	[277]
	*Glu-D1*	AS-PCR, KASP	Dx2, 3, 5, Dy10, 12	Various markers	[277]
LMW-GS	*Glu-A3*	STS, KASP	*a*, *b*, *c*, *d*, *e*, *f*, *g*	Various markers	[60,278]
	*Glu-B3*	AS-PCR, STS, KASP	*a*, *b*, *c*, *d*, *e*, *f*, *g*, *h*, *i*	Various markers	[60,279]
	*Glu-D3*	STS	*a*, *b*, *c*, *d*, *e*, *g*, *h*, *i*, *j*, *k*	Various markers	[118]
Starch properties	Wx-A1	SSR, RFLP, KASP, STS	Wx-A1a, b, c, d, e, f, g, h, i	SM126	[124,280]
	Wx-B1	SSR, STS	Wx-B1a, b, e	Various markers	[123]
	Wx-D1	SSR	Wx-D1a, b	F_2_-lines	[124]

## Data Availability

Not applicable.

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
