# Peer review of "The Role of *Puroindoline*, *Gpc-B1*, *Starch Synthase* Genes, and Gluten Proteins in Regulating End-Use Quality in Wheat"

_ijms, 2025, doi:10.3390/ijms26178565_

Round 1

Reviewer 1 Report

Comments and Suggestions for Authors

IJMS-3836207

This study reviewed the roles of puroindoline, Gpc-B1, starch synthase genes, and gluten proteins in regulation end-use quality in wheat.  The review focused on the progress, challenges, and opportunities in breeding for end-use quality in wheat.  It provides a comprehensive insight into the effects of genes in regulation end-use quality and serves as a baseline information for wheat breeders. However, several major and minor issues need to be addressed before publication.

Major issues:

  1. The subtitle of 3.1 is “The influence of Gpc-B1 genes on the end-use quality.” However, this section addresses not only the influence of Gpc-B1 genes on protein content but also on Fe and Zn concentrations. Therefore, I suggest revising this subtitle to “The influence of Gpc-B1 genes on protein content and nutritional quality.”
  2. Section 4 should also mention the basic conflict between yield and grain protein content (“dilution effect”). This is a key challenge for wheat breeding.
  3. In the introduction, the paper mentions that traditional quality phenotyping is costly and time-consuming, which is a core challenge in breeding. However, Section 5 (“Opportunities”) does not mention high-throughput phenotyping (Phenomics) as a potential solution. The authors should address this issue.

Minor issues:

  1. Line 27: “the effects of genes in regulating end-use quality” should be revised to “the effects of genes on regulating end-use quality”.
  2. Line 28: “a baseline information”. The word information is an uncountable noun and cannot be used with the indefinite article “a”. Therefore, it should be corrected to “baseline information”.
  3. Lines 420-421: In the sentence “Quality deterioration is one of the major attributes negatively affecting end-use quality traits and thus require consideration during the improvement of wheat cultivars”, the word “require” should be revised to “requires”.
  4. Lines 433-435: In the sentence “Stripe rust significantly reduces flour yield (FY) and increase flour ash content (FAC) due to shrivelled seeds, consequently affecting seed quality and grain yield”, the word “increase” should be corrected to the singular form “increases”.
  5. Line 447: The word “gene/s” should preferably be revised to “gene or genes”.
  6. Line 541: “End-use” should be changed to lowercase “end-use”.
  7. Line 557: “has been” should be revised to “have been”.
  8. Line 622: “have been located” should be corrected to “has been located”.
  9. In the Table 3, “Lillemo and Morris, (2000)” should be changed to “(Lillemo and Morris, 2000)”.
  10. Line 629: “Genome editing technologies refers to the process of deleting” should be revised to “Genome editing technologies refer to the process of deleting”.

Author Response

This study reviewed the roles of puroindoline, Gpc-B1, starch synthase genes, and gluten proteins in regulation end-use quality in wheat.  The review focused on the progress, challenges, and opportunities in breeding for end-use quality in wheat.  It provides a comprehensive insight into the effects of genes in regulation end-use quality and serves as a baseline information for wheat breeders.

Response: Thank for very much for dedicating time to review this manuscript and for your comments. Kindly find the detailed responses below and the corresponding revisions in track changes in the re-submitted files.

Major issues:

Comment 1: The subtitle of 3.1 is “The influence of Gpc-B1 genes on the end-use quality.” However, this section addresses not only the influence of Gpc-B1 genes on protein content but also on Fe and Zn concentrations. Therefore, I suggest revising this subtitle to “The influence of Gpc-B1 genes on protein content and nutritional quality.”

Response 1: Thank you for pointing this out. We agree with the comment. Therefore, we have revised the subtitle of 3.1. to “The influence of Gpc-B1 genes on grain protein content and nutritional quality”.

Comment 2: Section 4 should also mention the basic conflict between yield and grain protein content (“dilution effect”). This is a key challenge for wheat breeding.

Response 2: Agreed. We have included information on the dilution effect between grain yield and grain protein content in Section 4.

Comment 3: In the introduction, the paper mentions that traditional quality phenotyping is costly and time-consuming, which is a core challenge in breeding. However, Section 5 (“Opportunities”) does not mention high-throughput phenotyping (Phenomics) as a potential solution. The authors should address this issue.

Response 3: Thank you for the recommendation. We agree that it will add value to the manuscript. Therefore, we added information (in Section 5, lines 655-662) on high-throughput phenotyping as a potential solution to addressing the high costs of labor and time associated with traditional phenotyping.

Minor issues:

Comment 4: Line 27: “the effects of genes in regulating end-use quality” should be revised to “the effects of genes on regulating end-use quality”.

Response 4: Thank you for the comment. We have revised the word to “on”.

Comment 5: Line 28: “a baseline information”. The word information is an uncountable noun and cannot be used with the indefinite article “a”. Therefore, it should be corrected to “baseline information”.

Response 5: We agree with the comment, and we have removed the indefinite article “a”.

Comment 6: Lines 420-421: In the sentence “Quality deterioration is one of the major attributes negatively affecting end-use quality traits and thus require consideration during the improvement of wheat cultivars”, the word “require” should be revised to “requires”.

Response 6: Thank you for the comment. We have revised the word to “requires”.

Comment 7: Lines 433-435: In the sentence “Stripe rust significantly reduces flour yield (FY) and increase flour ash content (FAC) due to shrivelled seeds, consequently affecting seed quality and grain yield”, the word “increase” should be corrected to the singular form “increases”.

Response 7: Agreed. We have corrected the word to the singular form “increases”.

Comment 8: Line 447: The word “gene/s” should preferably be revised to “gene or genes”.

Response 8: Agreed. We have revised the word to “genes”.

Comment 9: Line 541: “End-use” should be changed to lowercase “end-use”.

Response 9: Thank you for pointing this out. We agree with the comment. Therefore, we have changed the word “end-use” to a lowercase.

Comment 10: Line 557: “has been” should be revised to “have been”.

Response 10: Thank you for the comment. We have revised the word to “have been”.

Comment 11: Line 622: “have been located” should be corrected to “has been located”.

Response 11: Thank you for pointing this out. However, we disagree with the comment because we used the word “QTL” in this context as a plural for the quantitative trait loci located in all 21 wheat chromosomes. Therefore, we cannot change the word “have” to a singular form “has”.

Comment 12: In the Table 3, “Lillemo and Morris, (2000)” should be changed to “(Lillemo and Morris, 2000)”.

Response 12: Thank you for the comment. We have changed the in-text citations to numbers according to the journal style of referencing.

Comment 13: Line 629: “Genome editing technologies refers to the process of deleting” should be revised to “Genome editing technologies refer to the process of deleting”.

Response 13: Agreed. We have revised the word to “refer”.

Reviewer 2 Report

Comments and Suggestions for Authors

In the manuscript, the authors highlight the technological progress, challenges, and opportunities in breeding for end-use quality in wheat. The manuscript is well written and reader-friendly. I suggest that the authors should add a column describing impacts on end-use quality units by the listed genetic modifications in both Table 1 and 3. Otherwise, I find the manuscript suitable for publication in the current special issue on IJMS.

Author Response

Thank for very much for dedicating time to review this manuscript and for your comment. Kindly find the detailed response below and the corresponding revision in track changes in the re-submitted files.

Comment 1: In the manuscript, the authors highlight the technological progress, challenges, and opportunities in breeding for end-use quality in wheat. The manuscript is well written and reader-friendly. I suggest that the authors should add a column describing impacts on end-use quality units by the listed genetic modifications in both Table 1 and 3. Otherwise, I find the manuscript suitable for publication in the current special issue on IJMS.

Response 1: In line 487-498, we highlighted the impact of transferring the 1RS.1BL, which introduces Sec-1 locus that is responsible for weakening the gluten and reducing the tolerance of dough to overmixing. We also mentioned that Sec-1 locus carries the γ- and ω-secalin proteins which are the main cause of celiac disease. In addition, we have referenced the table (in line 489) to link the two sections. The impact of 1Ay21* introgression was described in line 590–603. We have also added information about the impact of translocating 1RS.1BL on the grain protein content, starch, and rheological parameters in line 603–608.

Round 2

Reviewer 1 Report

Comments and Suggestions for Authors

I have reviewed the author's revisions and their response to my previous comments. The author has addressed all points raised.

The manuscript is now significantly improved, and I have no further concerns. I recommend it be accepted for publication in its current form.